



# Critical transitions in the hydrological system: Early-warning signals and network analysis

Xueli Yang[1], Zhi-Hua Wang[1], Chenghao Wang[2]

[1]School of Sustainable Engineering and the Built Environment, Arizona State University, Tempe, AZ 85287, USA

[2]Department of Earth System Science, Stanford University, Stanford, CA 94305, USA

*Correspondence to*: Zhi-Hua Wang (zhwang@asu.edu)

**Abstract.** In this study, we identified the critical transitions of hydrological processes including precipitation and potential evapotranspiration by analysing their early-warning signals and system-based network structures. The statistical early-warning signals are manifest in increasing trends of autocorrelation and variance in the hydrology system ranging from

regional to global scales, prior to climate shifts in the 1970s and 1990s in agreement with observations. We further extended the conventional statistics-based measures of early-warning signals to system-based network analysis in urban areas across the contiguous United States. The topology of urban precipitation network features hub-periphery (clustering) and modular organization, with strong intra-regional connectivity and inter-regional gateways (teleconnection). We found that several network parameters (mean correlation coefficient, density, and clustering coefficient) gradually increased prior to the critical

transition in the 1990s, signifying the enhanced synchronization among urban precipitation pattern. These topological parameters not only can serve as novel system-based early-warning signals to critical transitions in hydrological processes, but also shed new lights on structure-dynamic interactions in the complex hydrological system.

## 1 Introduction

The hydrological cycle plays an important role in the changing Earth's climate system, especially via the exchanges of heat

and moisture between the atmosphere and the Earth's surface (Chahine, 1992; Held and Soden, 2006; Oki and Kanae, 2006). However, as compared to temperature shifts, changes in global hydrological cycle (e.g. precipitation) are relatively less well-understood, despite the strong coupling between energy and water transport (Allen and Ingram, 2002; Marvel and Bonfils, 2013; Yang et al., 2019). Andrews et al. (2010) pointed out that the precipitation response to climate change can be roughly split into a fast response part strongly correlated with radiative forcing absorbed by the atmosphere and a relatively slow

response to global surface temperature change. Existing studies also showed that the change of global precipitation can be attributed to both natural changes (e.g., solar–volcanic forcing) and anthropogenic forcing (e.g., emission of greenhouse gases) (Liu et al., 2013; Marvel and Bonfils, 2013). A comprehensive review by Dore (2005) suggested increased precipitation in high latitudes of the Northern Hemisphere, decreased precipitation in China, Australia, and the Small Island States in the Pacific, and increased variance of precipitation in equatorial regions. According to IPCC (2014), annual



precipitation over the mid-latitude land areas of the Northern Hemisphere on average has increased since 1901, with the increasing number of heavy precipitation events in some regions. In addition, the Coupled Model Intercomparison Project Phase 5 (CMIP5) models (IPCC, 2014) predicted a nonuniform change of precipitation in the future. For example, mean precipitation will likely decrease in many mid-latitude and subtropical dry regions, whereas an increase in mean precipitation is projected in many mid-latitude wet regions under the RCP8.5 scenario.

Overall, precipitation in the contiguous United States (CONUS) has increased since 1990 with substantial seasonal and regional variations, and the projected precipitation changes over this century are not uniform (Melillo et al., 2014). For instance, Insaf et al. (2013) observed an increasing trend in annual precipitation over the New York State, with significant positive trend in several precipitation indicators (e.g., the number of heavy precipitation days and consecutive wet days) from 1948 to 2008. Similar wetting trends were observed in Brown et al. (2010) over the entire northeastern CONUS.

Existing studies have identified the connections between CONUS precipitation and some climate indices (especially those related to sea surface temperature anomalies), highlighting the importance role that climate variability plays in changing regional rainfall patterns (e.g., Barlow et al., 2001; Miller et al., 1994). Gutzler et al. (2002) found that the Southwest winter precipitation anomalies are strongly affected by the El Niño–Southern Oscillation (ENSO) cycle and the phase of the Pacific decadal oscillation using long-term (1950–1997) index analysis. In particular, the shift between dry and wet periods is tied to

the phase change of the Interdecadal Pacific Oscillation (IPO), as observed in many studies (e.g., Dai, 2013; Deser et al., 2004). For example, the switch of IPO from a cold phase to a warm phase around 1977 induced a clear upward trend in precipitation over much of the West and central part (e.g., Oklahoma, Kansas, Missouri) of the CONUS, whereas a latter shift in around 1999 (back to a cold phase) resulted in decreased precipitation (Dai, 2013). Owing to the close interactions among hydrological cycle (especially precipitation), ecosystems, and human society, catastrophic transitions in the

hydrological system impose severe risks for people, economies, and ecosystems, and increase their vulnerability to water shortage, storms, flooding, and drought (Melillo et al., 2014). Future climate changes, along with population growth, will further amplify these existing risks in many regions (IPCC, 2014). To adjust existing mitigation and adaptation policies and proactively develop new strategies, it is imperative to identify critical hydrological transitions and their interplay with climate system dynamics.

Existing research has identified critical transitions (aka tipping points) in various dynamic systems, during which the system shifts from one state to another induced by small perturbations (Scheffer et al., 2009; Lenton, 2013). As approaching a critical threshold, a dynamic system will slow down in recovering from small perturbations; this phenomenon is also known as critical *slowing down* (Litt et al., 2001; Van Nes and Scheffer, 2007; Venegas et al., 2005). Intuitively, the intrinsic rate of change in a system decreases with increasing memory for perturbations (Scheffer et al., 2009). Mathematically, the

maximum real part of the eigenvalues of the Jacobian matrix tends to zero in the critical slowing down, often with *early-warning signals* such as increasing autocorrelation, return time, skewness, and variance (Lenton, 2011; Ives, 1995; Carpenter and Brock, 2006; Scheffer et al., 2009). In particular, early-warning signals have been identified for abrupt climate change based on paleoclimatic records and numerical simulations (Dakos et al., 2008; Lenton, 2011). For example, Dakos et al.





(2008) found that eight ancient abrupt climate shifts were all preceded by a slowing down of the fluctuations (with early-warning signals of increased autocorrelation) before the actual shift. It is noteworthy that these studies usually have time scales ranging from $10^4$ to $10^6$ years, with relatively few implications for concurrent climate and variability in the Anthropocene. Recently, Wang et al. (2020b) identified early-warning signals in the early 20[th] century global warming and heat waves with time scales much shorter than existing paleoclimate studies. Nevertheless, the existence of early-warning signals in hydrological system remains largely obscure.

On the other hand, interactions among hydrological processes and climate variability reveals that the complex system dynamics run on top of the topological substratum of connected players (nodes), viz. a *network*. Over recent years, more research effort has been devoted to applications of the network theory to hydrological and climate systems (e.g. Boers et al., 2016, 2019; Fan et al., 2017; Konapala and Mishra, 2017; Wang and Wang, 2020). For instance, Konapala and Mishra (2017) analysed the spatiotemporal propagation of droughts in the CONUS based on three network-based metrics (strength, direction, and distance). Complex network theory has also been employed to investigate regional and global nonlinear and long-range connections (teleconnections) for different types of rainfall events as well as the synchronization of extreme rainfall events (Boers et al., 2016, 2019; Rheinwalt et al., 2016). In particular, the change in hydrologic cycle may be reflected by the variations in the topological structure of networks. The climate shift in the mid-1970s, for instance, has been investigated via the coupling strength of the network based on major climate indices (Tsonis et al., 2007). Tsonis et al. (2008) also showed that the "supernodes" (nodes connecting with many other nodes) in the climate network correspond to major atmospheric teleconnections.

Despite being a powerful new tool, the application of network analysis to urban climates, hydrological processes in particular, remains much under-explored up to date. Today, urban areas accommodate more than half of the global population and serve as the engine of socio-economic development. Meanwhile, cites feature concentrated anthropogenic activities and emissions of waste heat, pollutants, and greenhouse gases, and hence confront more severe climate extremes and environmental issues via a cascade of land-atmosphere interactions (Pielke et al., 2007; Song and Wang, 2015, 2016). Few studies have examined the climate similarity and connectivity of CONUS cities. Fitzpatrick and Dunn (2019) quantified the similarity of future climate to contemporary climate of cities in North American using the sigma dissimilarity approach, and visualized climate analogues of future cities. Wang et al. (2020a) identified urban clustering patterns in response to environmental stressors (precipitation, surface temperature, and aerosol optical depth) with different temporal scales based on affinity propagation method.

In this paper, we aim to investigate critical transitions in hydrological processes, primarily precipitation at regional and global scales. The remainder the paper is organized as follows: we present data sources of precipitation and potential evapotranspiration (PET) in Section 2, together with definition of early-warning signals and basic network analysis techniques. These methods are then applied to urban areas in CONUS with results presented in Section 3: statistical variance and auto-correlation in Section 3.1, and changes in network structure in Section 3.2. We conclude this study with main findings and future perspectives in Section 4.



## 2 Methods

### 2.1 Data sources

In this study, we analyse the statistical and topological measures of abrupt precipitation and PET changes at multiple scales, ranging from regional to global scales. For global scale analysis of early-warning signals, we retrieved the long-term (1901–2018) gridded global monthly precipitation and PET data with a spatial resolution of 0.5° × 0.5° from the University of East Anglia Climatic Research Unit Time-Series (CRU TS) dataset version 4.03 (http://data.ceda.ac.uk/badc/cru/data/cru_ts/cru_ts_4.03/data). This gridded dataset covers all land domains of the world
except Antarctica (Harris et al., 2020). CRU TS was produced using the angular-distance weighting (ADW) method to interpolate monthly climate anomalies based on extensive weather station observations. Here we calculated the annual global means of precipitation and PET over land as the weighted averages of the Northern Hemisphere and Southern Hemisphere following Osborn and Jones (2014). Note that means for each hemisphere are the areal-weighted averages of all non-missing values.

For regional scale analysis of early-warning signals, we obtained annual precipitation time series for selected cities in the CONUS from the NOAA National Centers for Environmental Information's Climate at a Glance database (https://www.ncdc.noaa.gov/cag/city/time-series). This city-level database contains monthly temperature and precipitation data for 215 U.S. cities, among which 27 cities have data recorded by Automated Surface Observing System (ASOS) stations, and the remaining 188 cities use Global Summary of the Month (GSOM) data. In particular, the GSOM dataset is
based on the Summary of the Day observations of the Global Historical Climatology Network-Daily (GHCN-Daily) dataset (Menne et al., 2012), in which the total monthly precipitation is based on daily or multi-day (if daily is missing) precipitation report. GHCN-Daily data have been quality controlled using a suite of automated algorithms designed to detect as many errors as possible meanwhile with a low false-positive rate (valid observations erroneously identified as invalid) (Durre et al., 2010). Additional quality control has also been performed with a validation process that involves independent
calculations and cross-comparisons to ensure computational accuracy.

For the subsequent network analysis, we focus on precipitation over all cities in the CONUS. We retrieved the 1 km × 1 km monthly precipitation data over the CONUS (1980–2018) from Daymet Version 3 dataset (Thornton et al., 2018). This dataset uses spatial convolution of a truncated Gaussian weighting filter applied to a number of stations (on average 15 for precipitation) (Thornton et al., 1997). Daymet uses daily precipitation measured by ground-based meteorological stations
from the GHCN-Daily dataset (Menne et al., 2012). Different from the regional scale analysis of early-warning signals (single station for each city), here we derived the time series of monthly precipitation spatially averaged over each urban area in the CONUS based on Daymet dataset. Note that urban areas (or cities) in this study are defined as areas with densely developed land and over 50,000 population, and the city boundaries are retrieved from the Topologically Integrated Geographic Encoding and Referencing (TIGER) system, U.S. Census Bureau
(https://www.census.gov/geographies/mapping-files/time-series/geo/tiger-geodatabase-file.html).



## 2.2 Early-warning signals for critical transition

### 2.2.1 The characteristic changes of critical slowing down

The critical slowing down near the catastrophic transition can be related to many statistical measures of the system dynamics subject to perturbations. The most widely used two measures are variance (or standard deviation; s.d.) and autocorrelation, both expected to increase near the catastrophic transition. This can be expressed with a simple autoregression model with lag-1 ($AR_1$) of the perturbation (Scheffer et al., 2009; Wang et al., 2020b),

$$\varepsilon(t_{n+1}) = \alpha\varepsilon(t_n) + \sigma R_n , \tag{1}$$

where $\varepsilon(t_n)$ is the deviation of the state variable (e.g., precipitation or PET in this study) from the equilibrium at time $t_n$, $R_n$ is a random number sampled from a normal distribution with the standard deviation of $\sigma$, and the autocorrelation $\alpha = e^{\lambda\Delta t}$ with $\lambda$ the rate of recovery from the perturbation (zero for white noise and one for red noise). The statistical expectation of this autocorrelation process is therefore (Ives, 1995)

$$\mathbf{E}(\varepsilon) = \frac{1}{1-\alpha} , \tag{2}$$

and the variance is given by

$$\mathrm{Var}(\varepsilon) = \frac{\sigma^2}{1-\alpha^2} . \tag{3}$$

The recovery speed $\lambda$ approaches zero as the return speed to equilibrium decreases (i.e., slows down) when a system close to the critical transition. As a result, the autocorrelation coefficient increases to one and the variance (or standard deviation) evolves toward infinity (Scheffer et al., 2009). The increase in the autocorrelation and variance (or standard deviation) of the fluctuations due to critical slowing down near transitions could serve as early-warning signals for the catastrophic threshold in the system.

### 2.2.2 Statistic measures of early-warning signals for critical transition

To quantify the early-warning signals presaging the system slowing down, we first identified the time instant (year) of critical transition in the precipitation and PET time series. More specifically, we first calculated the cumulative precipitation and PET time series (not shown here) based on original time series. The year of critical transition was determined based on the abrupt change of slopes in each cumulative time series. We then divided each original precipitation (or PET) time series into two (quasi)stationary parts using this critical transition year. The anomalies of PET and precipitation were derived by subtracting from the time series their mean values in the periods prior and posterior to the transition correspondingly. The subsequent analyses focus on the parts prior to the occurrence of critical transitions, whereas the parts posterior to the transitions are truncated as irrelevant to early warning.





The early-warning signals for global and city-scale changes are quantified using $AR_1$ and s.d.; the mechanism is detailed in

Section 2.2.1 above (Ives, 1995; Carpenter and Brock, 2006; Van Nes and Scheffer, 2007). In addition, 7 and 13 years are

selected as the sliding window sizes $w$ for early-warning signal analysis of short-term and long-term data series, respectively,

based on a previous sensitivity analysis (Tsonis et al., 2007). Sliding windows ensure sufficient number of samples in

estimating correlations and standard deviations, meanwhile avoiding the dilution of signals in critical transitions (Wang et

al., 2020b).

The lag-1 autocorrelation, $AR_1$, with sliding window size $w$ centred at $x_k$ is given by

$$
AR_{1,k} = \frac{\sum_{i=k-(w-1)/2-1}^{k+(w-1)/2-1} [(x_i - \mu_{t_1})(x_{i+1} - \mu_{t_2})]}{\left[\sum_{i=k-(w-1)/2-1}^{k+(w-1)/2-1} (x_i - \mu_{t_1})^2\right]^{1/2} \left[\sum_{i=k-(w-1)/2-1}^{k+(w-1)/2-1} (x_{i+1} - \mu_{t_2})^2\right]^{1/2}} ,
\tag{4}
$$

where $x_i$ denotes the variables (annual precipitation and PET anomalies), $\mu_{t_1}$ and $\mu_{t_2}$ are arithmetic averages of variables in

the intervals $[k - (w - 1)/2 - 1, k + (w - 1)/2 - 1]$ and $[k - (w - 1)/2, k + (w - 1)/2]$, respectively. The (sample) standard

deviation (s.d.$_k$) is computed as (Wang et al., 2020b)

$$
\text{s.d.}_k = \sqrt{\frac{1}{w-1} \sum_{i=k-(w-1)/2}^{k+(w-1)/2} (x_i - \mu_{t2})^2} .
\tag{5}
$$

Next, we illustrate the characteristic changes of $AR_1$ and s.d. in an autocorrelation process using a sample model of harvested

population (Beddington and May, 1977). Its governing dynamics is given by a stochastic differential equation

$$
\frac{dX}{dt} = [r(t) - E]X - r_0 X^2 / K ,
\tag{6}
$$

where $dX/dt$ is the net growth rate of population $X$; $r(t) = r_0 + \gamma(t)$ with $r_0$ the mean value of intrinsic growth rate (set to 0.6 in

this example), and $\gamma(t)$ a white noise with zero mean; $K$ is the carrying capacity (set to 10); and $E$ is the harvesting rate.

Figure 1 shows the evolution of population subject to the dynamics in Eq. (6) and results of two statistical metrics with

different harvesting rates (0.1 and 0.4 for low and high rates, respectively). When the system is far from the tipping point

(low harvesting rate; Fig. 1a and b), its resilience to perturbations is large with a relatively high recovery rate, characterized

by relatively low $AR_1$ and s.d. In comparison, the resilience decreases when the system is closer to the critical transition

(high harvesting rate; Fig. 1c and d), and the rate of recovery from perturbations declines as a consequence of critical

slowing down. The system has a relatively long memory for perturbations, resulting in a stronger correlation between

subsequent states and a larger standard deviation in a stochastic environment (Scheffer et al., 2009).



## 2.3 Precipitation network analysis

We have illustrated how $AR_1$ and variance of critical transitions of precipitation and other hydrological processes evolve
when the hydrological systems approach critical transitions. These statistical measures follow the system dynamics
represented by, e.g., an autocorrelation process, and are applicable to time series at various spatial scales. In addition, the
hydrological system dynamics often evolve on top of complex topological structures such as climate networks (Konapala
and Mishra, 2017); their interactions modulate the potential tipping of the system. Nevertheless, it remains obscure up to
date how critical transitions in hydrological systems emerges and become representative in their corresponding network
structure. In this section, we extend the analysis of early-warning signals for precipitation transitions beyond the changes of
conventional means (viz. $AR_1$ and variance), and investigate the system structure represented by the precipitation network of
all CONUS cities.

In its simplest form, a network (or graph) can be mathematically represented as a group of nodes or vertices that are
connected together. The connection between a pair of nodes is called a link (edge), representing the similarity (or inverse
distance) of attribute of the two nodes. Note that here we focus on undirected and unweighted networks only, while the
proposed procedure of analysis in this study remains applicable to more complicated (e.g. directed or/and weighted)
networks with slight modification.

More specifically, in this study, we treat 481 CONUS cities (see Section 2.1) as nodes and construct the network based on
monthly precipitation retrieved from the Daymet dataset (Thornton et al., 2018) for the period of 1980–2018 (see Section
2.1). The connectivity of precipitation networks, are constructed using the commonly adopted distance function, viz. the
Pearson correlation coefficient $\rho_{ij}$, between monthly precipitation time series of two cities $i$ and $j$, defined as

$$\rho_{ij} = \frac{\mathbf{E}\left[(P_i - \mu_{P_i})(P_j - \mu_{P_j})\right]}{\sigma_{P_i}\sigma_{P_j}}, \tag{7}$$

where $P_i$ is the precipitation time series with the subscript $i$ denoting the node (city) number; $\mu$ and $\sigma$ are the mean and
standard deviation of the precipitation time series, respectively. Note that the correlation coefficient is only one kind of
measures to describe connectivity, while other similarity (or dissimilarity) functions (e.g., Minkowski distance) (Wang et al.,
2020a) are also applicable in network construction. The connectivity between a pair of nodes forms the adjacency matrix $A_{ij}$,
$i, j = 1, 2, …, N$, defined by (Tsonis and Roebber, 2004),

$$A_{ij} = \Theta(\rho_{ij} - \rho_{\text{threshold}}), \tag{8}$$

where $\Theta$ is the Heaviside step function, $\rho_{\text{threshold}}$ is the threshold value, and $N$ is the number of cities. Here we choose a
threshold of 0.5, a statistically significant example as suggested by previous studies (Tsonis and Roebber., 2004; Wang and
Wang, 2020).





To investigate the topological feature of the precipitation network and how it evolves over time, four representative metrics are then analysed, viz. the mean distance (or equivalently the network-average Pearson correlation coefficient), the density, the network modularity, and the clustering coefficient. The mean distance describes the overall connectivity of the network, and is given by (Tsonis et al., 2007),

$$d(t) = \frac{2}{N(N-1)} \sum_{d_{ij}^t \in D^t} d_{ij}^t \, ,$$

(9)

where $d(t)$ denotes mean network distance at current time $t$ at the centre of a sliding window; $d_{ij}^t$ represents the distance between a pair nodes $i$ and $j$ at time $t$; and $D^t$ is the distance matrix for each sliding window. Equivalently, the mean network distance can be measured using the Pearson correlation coefficient, as

$$d_{ij}^t = \sqrt{2(1-|\rho_{ij}^t|)} \, .$$

(10)

The distance can be thought as the average correlation between all possible pairs of nodes and is interpreted as a measure of synchronization among different components of a network: a distance of zero corresponds to a complete synchronization (Tsonis et al., 2007). The mean Pearson correlation coefficient is

$$\rho(t) = \frac{2}{N(N-1)} \sum_{\rho_{ij}^t \in D^t} \rho_{ij}^t \, ,$$

(11)

and it is clear that the mean distance and the mean Pearson correlation coefficient are inversely correlated when the latter is positive.

The density of a network is the fraction of edges that are present with the value lies between 0 and 1, calculated by,

$$\rho' = \frac{2m}{N(N-1)} \, ,$$

(12)

where $m$ is the total number of edges. The density represents the probability that a pair of nodes picked at random from the whole network is connected by an edge. It plays an important role in the random graph model (Newman 2018). The larger value of the density, the denser the network is.

The modularity is the number of edges falling within groups minus the expected number in an equivalent network with edges placed at random. It describes how strong the community structure is (Newman et al., 2006), and is defined by,

$$Q = \frac{1}{2m} \sum_{ij} (A_{ij} - \frac{k_i k_j}{2m}) \delta_{gigj} \, ,$$

(13)



where $g_i$ is the group to which node $i$ belong, $\delta_{ij}$ is the Kronecker delta function, and $k_i$ is the degree (number of links) of node $i$. The network with positive and large values of modularity is preferable on research and real world (Newman and Girvan, 2004).

The clustering coefficient of a network is defined as,

$$C = \frac{1}{N} \sum_{i=1}^{N} \frac{n_i}{k_i(k_i - 1)/2},$$ (14)

where $n_i$ is the number of edges among the nearest neighbours of the $i$-th node. Topologically, the clustering coefficient is the fraction of paths of length two in the network that are closed and quantifies the extent to which pair of nodes with common neighbour are also neighbours of each other (Newman, 2018).

## 3 Results and discussion

### 3.1 Statistical measures of early-warning signals

We first identify the conventional early-warning signals at the global scale, viz. $AR_1$ and s.d., for the potential evapotranspiration and precipitation anomalies using the algorithm detailed in Section 2.2. Figure 2 shows PET and precipitation anomalies and the early-warning signals prior to critical transitions. The critical transition of global PET occurred in year 1994 (denoted by the red vertical line in Fig. 2a), which is in general consistent with the transition of solar radiation trend at Earth's surface in ~1990 (Pinker et al., 2005; Wild et al., 2005). This transition from decreasing to
increasing global radiation (also known as the transition from global dimming to brightening) has been found in many observational records, primarily due to the changes in cloudiness, aerosol loadings, and atmospheric transparency (Pinker et al., 2005). It is clear from Fig. 2a that both $AR_1$ and s.d. of global PET increased gradually over time for more than one decade (one sliding window) prior to the critical transition. Per theoretical analysis in Section 2, the increase in $AR_1$ and s.d. apparently presages the critical slowing down of the rate of recovery from perturbation, as the system evolved approaching
the transition in 1994.

In addition, we observe similar increasing trends in the time evolution of $AR_1$ and s.d. for global precipitation anomalies, where the critical transition happened in 1979. The time of critical transition is in good agreement with the climate shift associated with IPO phase change from cold to warm around year 1977, with significant influences on precipitation (especially over the CONUS) (Dai, 2013; Deser et al., 2004). It is noteworthy that this critical transition in global
precipitation due to the phase change of IPO is relatively stronger than a later transition in ~1999 when IPO phase shifted back to cold state (Dai, 2012).

We then proceed to identify early-warning signals for precipitation at city scale, following the same procedure for the global scale analysis above. Figure 3 shows the results of four CONUS cities (New York City, NY, Seattle, WA, Fresno, CA, and Miami, FL), each with distinct background climates. Note that for New York City and Fresno the monthly precipitation data



are from 1895 to 2019 (125 years), while for Seattle and Miami the data are from 1948 to 2019 (72 years). For individual
cities, the time (year) of critical transition occurred in precipitation records differ from one another: 1967 for New York City,
1993 for Seattle, 1971 for Fresno, and 1993 for Miami. The results are generally in good agreement with the studies on phase
change related to IPO from a cold to a warm around 1977, especially for the New York city and Fresno (Deser et al., 2004)
around 1970s. The phase change was associated with significant changes in global temperature and ENSO variability. For
Seattle and Miami, the transitions in the 1990s are in line with the transition of IPO phase from warm to cold (Dai, 2012).
The results of early-warning signals are also shown in Fig. 3. Here again we find that, in general, the statistical measures of
$AR_1$ and s.d. increase with time in all four cities prior to the emergence of transition.

**3.2 Network representation of critical transitions**

In this section, we extend the concept of early-warning signals for critical transitions in hydrological systems from
conventional time series analysis to the topological analysis of their network structure. We first construct the precipitation
networks based on the monthly precipitation data for all CONUS 481 cities in the period 1980-2018 retrieved from the
Daymet dataset (Thornton et al., 2018), with a threshold similarity of 0.5 (detailed in Section 2.3). The CONUS precipitation
network generated using the entire time series (39 years × 12 months) is shown in Fig. 4a. All CONUS cities are further
subdivided into nine geographic regions following Wang and Wang (2020), as listed in Table 1. The connectivity, viz. the
adjacency matrix, of the precipitation network is shown in Fig. 4b, with the 9 regions marked.
From Fig. 4, it is clear that the 39-year aggregated CONUS urban precipitation network is largely occupied by intra-regional
connection, manifest as regional clusters in the geographic map (Fig. 4a) and dark (connected) diagonal blocks in the
adjacency matrix (Fig. 4b). This is consistent with a clustering analysis based on monthly precipitation (1981–2010) in
CONUS cities (Wang et al., 2020a). This is physical as the patterns of precipitation and its anomalies in a city are
predominated by their geographic controls and background climate conditions in the region where the city is located. In
other words, the dense intra-regional connections (diagonal blocks in Fig. 4b) reveal the similarity of precipitation patterns
among cities with similar climate environment, viz. "like is connected to like" (Newman and Girvan, 2004). This topological
feature also shows that the CONUS precipitation network is highly assortative with large modularity $(Q = 0.589)$.
In addition to the modular structure reflected as dense intra-regional connectivity, the precipitation network also includes
some inter-regional (Fig 4a.) or "off-diagonal" (Fig. 4b) connections. The presence of long-range, out-of-region connectivity
is usually formed by complex interactions via atmospheric gateways (teleconnection) (Boer et al., 2019). For example, it is
found that precipitation patterns are similar in the Ohio Valley (Region 1) and Upper Midwest (Region 2) (same for West
and Northwest), shown as the off-diagonal connection in Fig. 4b. Similar hydrological teleconnection have been observed in
Konapala and Mishra (2017) and Wang et al. (2020). For example, Konapala and Mishra (2017) found that the Ohio Valley
region plays an important role in propagating droughts (with higher values of outward-strength) toward adjacent regions
such as Upper Midwest (Region 2) and Northeast (Region 3), consistent with off-diagonal connections shown in Fig. 4b.


To find how the network structure respond to (or presage, as early-warning signals of AR1 and s.d.) critical transitions in the CONUS precipitation system, we then proceed to construct the *time-varying* networks (cf. the aggregated 39-year network in Fig. 4) using a sliding window of 7 years (following Section 2.2) from the same set of Daymet precipitation data in the

period of 39 years (1980-2019). In addition, we follow the same procedure for analysing the time series of cumulative precipitation in CONUS and determine the year of critical transition at the emergence of abrupt change in the slope of cumulative precipitation. In this case, the critical transition for precipitation in CONUS urban areas was around the year 1998, which is consistent with the IPO phase changes in the 1990s (Dai, 2013; Deser et al., 2004). This can also be seen from Fig. 5a as that the early-warning signal of s.d. had a clear increasing trend, whereas the trend of $AR_1$ was slightly

obscure in the window size of 7 years. Nevertheless, a clear increase immediately prior to the transition is still observed for $AR_1$ in Fig. 5a.

In addition, we also find that the network topology has a significant response to the critical transition as shown in Fig. 5. The mean Pearson correlation coefficient $\rho$, the network density $\rho'$, and the clustering coefficient $C$ all exhibited clear trend of increase prior to the transition. Note that $\rho$ is the structural parameter *prior* to the construction of networks, whereas $\rho'$ and $C$

are *posterior* parameters determined from the topology of constructed networks. Also the mean distance $d$ is (nearly) inversely correlated to the mean Pearson correlation coefficient, so it is expected that $d$ decreased prior to the transition. The increase in $\rho$ (or equivalently, the decrease in $d$) suggests that on average the connectivity among all cities in the CONUS precipitation network was enhanced prior to the transition. This is likely attributable to the strengthening of synchronization of precipitation events (Boers et al., 2016; Rheinwalt et al., 2016) via, e.g., propagation of precipitation fronts entraining

larger urban areas than before, as a result of critical slowing down in recovery from system perturbation (e.g., extreme rainfalls). Similar patterns of increasing in structural correlation $\rho$ and decrease in mean distance have been observed in other climate networks by Tsonis et al. (2007).

The enhanced network connectivity prior to critical transitions, manifested as increase in nodal correlation $\rho$ (or decrease in mean distance $d$), in turn, led to the increase in the total number of links and the network density $\rho'$ (Fig. 5b). The clustering

coefficient $C$ also increased as a result of enhanced community structure predominated by the intra-regional connections (Fig. 4). In contrast, the modular structure of the precipitation network remained insusceptible to the emergence of critical transition and enhanced synchronization of precipitation events, with the value of modularity $Q$ fluctuating around 0.55~0.57 in the 7-year window size. This is seemingly because the critical transition tends to enhance both the local community structure (intra-regional connection) as well as the teleconnection (inter-regional connection), hence leaving the modular

organization of the entire network unchanged. Qualitatively, this means that the topological structure of the CONUS precipitation network, despite its temporal evolution and changes in the overlying dynamics (viz. the occurrence of critical transition or slowing down), remains "like is connected to like".



## 4 Concluding remarks

In this study, we investigated the early-warning signals for potential catastrophic transitions in the hydrological system, in
particular the precipitation and PET, as a result of the critical slowing down of system recovery rate from dynamic perturbation. The occurrence of critical transitions in precipitation patterns was in agreement with recorded observation, consistent with climate shifts in 1970s and 1990s due to low frequency variability such as SST or IPO. The theoretical basis of their early-warning signals, statistically measured in terms of $AR_1$ and the standard deviation (variance), was illustrated using an autocorrelation process and a harvested population model. We applied the analysis to the longterm global scale
precipitation and PET time series as well as city scale precipitation dataset in CONUS. It was found that the emergence of increasing trends in AR1 and s.d., prior to critical transitions in all cases, agrees well with theoretical predictions.

In addition, we extended the conventional statistical measures of early-warning signals to system-based network analysis. We constructed precipitation networks over all urban areas in CONUS and calculated some key topological parameters including the mean similarity/distance function, network density, clustering coefficient, and modularity. The system
evolution toward the critical transition apparently enhances the synchronization of precipitation events over CONUS urban areas, leading to the strengthening of the community structure as well as teleconnections in the network. As a result, we identified the increasing trends of the mean network correlation, the density, and the clustering coefficients due to system transitions; these structural parameters can therefore be used as network-based precipitation system early-warning signals, similar to the process-based measures of $AR_1$ and variance. The network modularity, on the other hand, is not susceptible to
and cannot be interpreted as a harbinger of critical transitions in the precipitation system. These findings help to shed new light on the intricate structure–dynamic interactions in complex climate systems that modulate the future trend of evolution of hydrological processes under global climate change.

## Competing interests

The authors declare that they have no conflict of interest.

## Acknowledgements

This work was supported by the U.S. National Science Foundation (NSF) under grant number AGS-1930629 and CBET-2028868, and the National Aeronautics and Space Administration (NASA) under grant 80NSSC20K1263.



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





**Figures:**

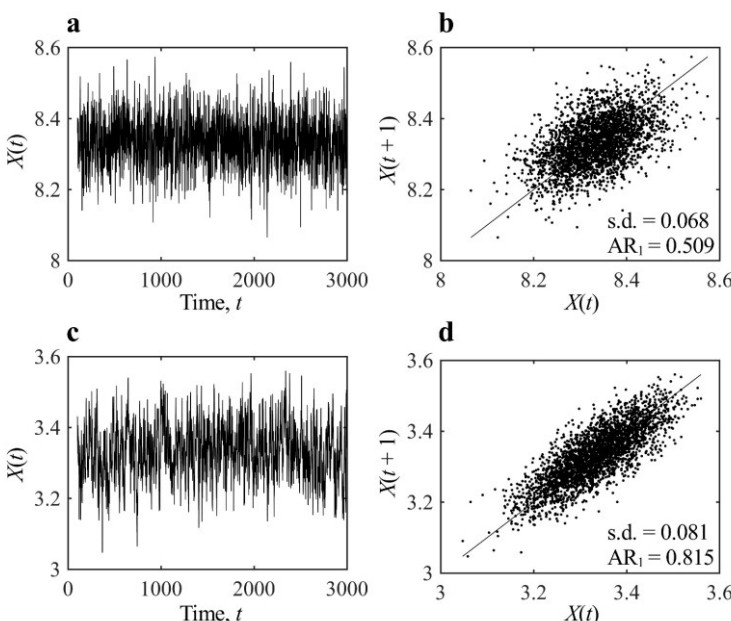

**Figure 1:** Characteristic changes in a system of harvested population when approaching the critical transition: (a) population dynamics and (b) statistical metrics when the system is far from tipping ($E = 0.1$); (c) and (d) the same as (a) and (b) but when the system is closer to tipping ($E = 0.4$). Note that the spin-up periods are removed.

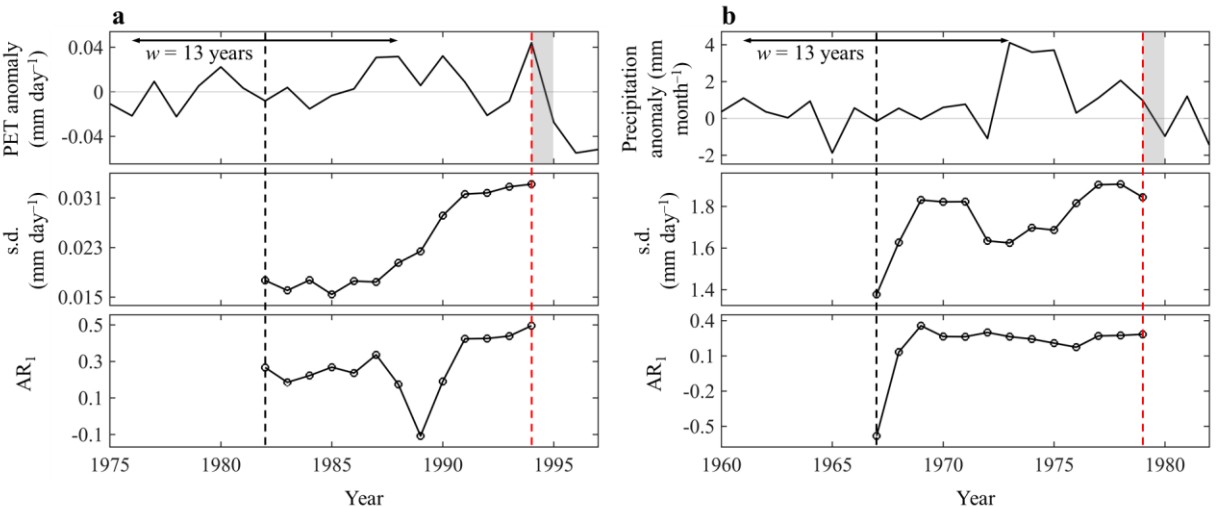

**Figure 2:** Time series of anomalies, $AR_1$, and s.d. of global (a) PET and (b) precipitation. Horizontal lines with arrow show the width of one moving window (13 years). Vertical dashed red lines represent critical transitions, and gray bands the transition phase.



**Figure 3:** Time series of anomalies, $AR_1$, and s.d. of precipitation in (a) New York, and (b) Seattle, (c) Fresno, and (d) Miami in CONUS. Horizontal lines with arrow show the width of one moving window (13 years). Vertical dashed red lines represent critical transitions, and gray bands the transition phase.



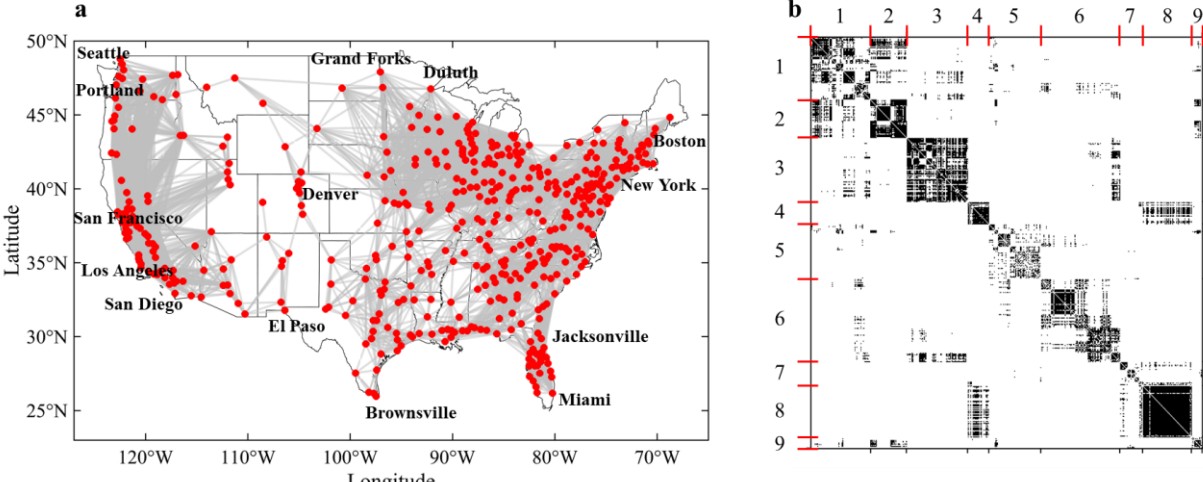

**Figure 4:** The precipitation network of CONUS cities: (a) the geographic map of connectivity and (b) the adjacency matrix, with $A_{ij} = 1$ in black (connected), $A_{ij} = 0$ in white, and cross short red line marking the division of nine geographic regions.


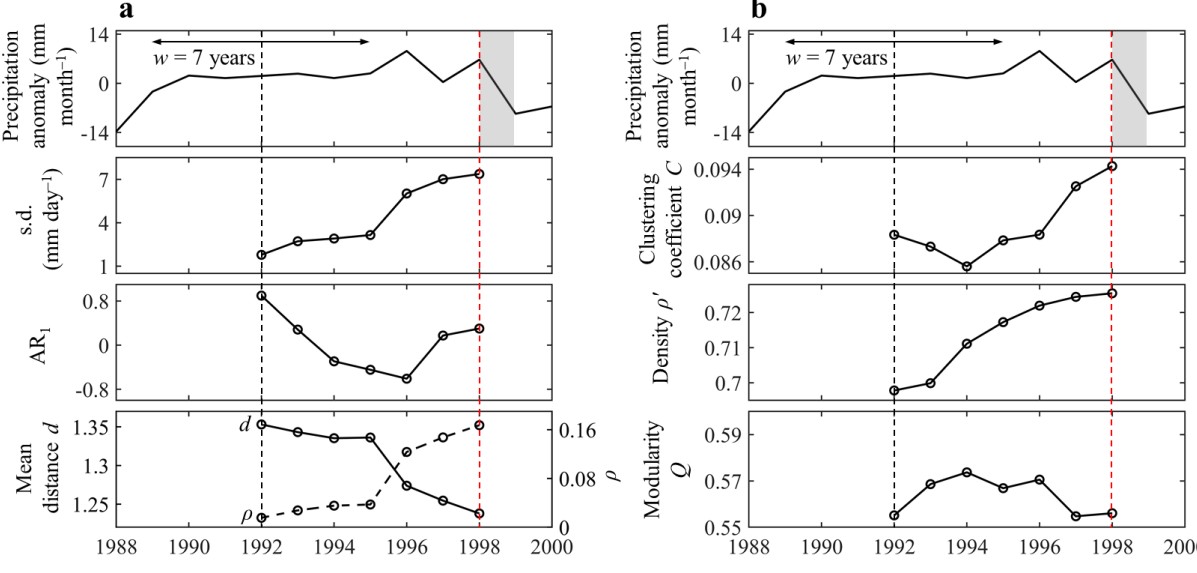

**Figure 5:** Early-warning signals and structural responses to the critical transitions (in year 1998) in time-varying CONUS precipitation, as the time evolutions of: (a) precipitation anomalies, $AR_1$, s.d., and the mean distance ($d$) and similarity ($\rho$) functions for network construction, and (b) network topological parameters of clustering coefficient ($C$), density ($\rho'$), and modularity ($Q$). Horizontal lines with
arrow show the width of one moving window (7 years). Vertical dashed red lines represent critical transitions, and gray bands the transition phase.



**Table:**

**Table 1:** The division of CONUS cities in nine geographical regions

| Number | Region name | States |
|---|---|---|
| 1 | Ohio Valley | IL, IN, KY, MO, OH, TN, and WV |
| 2 | Upper Midwest | IA, MI, MN, and WI |
| 3 | Northeast | CT, DE, MA, MD, ME, NH, NJ, NY, PA, RI, and VT |
| 4 | Northeast | ID, OR, and WA |
| 5 | South | AR, KS, LA, MS, OK, and TX |
| 6 | Southeast | AL, FL, GA, NC, SC, and VA |
| 7 | Southwest | AZ, CO, NM, and UT |
| 8 | West | CA and NV |
| 9 | Northern Rockies and Plains | MT, ND, NE, SD, and WY |
