# Peer review of "Critical transitions in the hydrological system: Early-warning signals and network analysis"

_Hydrology and Earth System Sciences, 2021_

## Author Comment (AC1)

**Response to the comments of Reviewer #1:**

*This paper applies the autocorrelation process and a harvested population model as well as network analysis for the early warning signals of transition in the hydrological system at the global and CONUS city scale.*
*Overall, the paper is well-written and certainly contains many novel ideas potentially helpful for the boarder community.*

We thank the reviewer for the constructive feedback and help in improving the quality of this manuscript. Below are detailed responses to the comments. All changes and clarifications were included in the revised manuscript.

*I see the paper will be strengthened by:*

*1. Line 50: important role*
We have the typo corrected.

*2. Line 219 – 222: discussion about the relative magnitudes of AR1 and s.d. I agree that visually figures 1b and d are quite distinct, but the authors may want to provide more information about whether any objective measures of high and low exist or any reference state exists in AR1 and s.d. in gauging the state of the system regarding how far away from the tipping point.*
The quantification of threshold values of AR1 and s.d. to determine how far the system is from the tipping point varies from case to case. The particular case presented in Fig. 1 only shows the increasing *trend* of AR1 and s.d., when the system is approaching the tipping point. The asymptotes of both measures, to the best of our knowledge, have not been worked out; it will be interesting, though challenging, to quantify the values of AR1 and s.d. at the tipping point by casting this practical problem in the analytical framework of Scheffer et al. (2009), e.g. Eqs. (2) & (3).

*3. In addition, it appears unclear to the reviewer whether Figure 1 is an example toy problem to illustrate the concepts or related to the main finding.*
It is true that the benchmark problem show in Figure 1 is not directly related to the subsequent applications to precipitation and PET in the study. Nevertheless, we think it is a good example for illustrating the concept of critical transition, especially the increasing trends of AR1 and s.d. Figure 1 is based on a classic harvest model in which the stability of population can be controlled by the parameter harvesting rate $E$. The increasing $AR_1$ and s.d. in population time series due to increasing $E$ as shown in Figure 1 signify that the population is approaching the tipping point, and these characteristics can serve as early-warning signals. Similar characteristics will be used in the following sections to determine how the hydrological systems evolve approaching critical transitions. We clarified this in the context.

*4. 2.3: the beginning few sentences seem to be repetitive of part of the introduction – therefore, may be better to combine with the introduction or shorten it.*
Thanks for the comment. We removed first two sentences in this section to make the presentation more concise.

*5. Line 253-254: single-plural mismatch*
Corrected in the revision.

*6. "The year of critical transition was determined based on the abrupt change of slopes in each cumulative time series. We then divided each original precipitation (or PET) time series into two (quasi)stationary parts using this critical transition year" – maybe good to provide in the appendix, since this is quite important. Also, may want to give more explanation for what you mean by the two parts being quasi-stationary.*
Thanks for the comment. Below we demonstrate in Fig. R1, using the example of Miami, how the year of transition (1998) was determined. We bisected the cumulative precipitation data (scatters) and fitted each segment using linear regression (thus each being *quasi-stationary* with a constant slope). The year of transition is determined as the intersection of two trend lines (solid red: prior to transition, and dash red: after transition) with different slopes. In addition, Fig. R1b shows the annual (not cumulative) precipitation, where the two different means (prior to and after the transition year) are subtracted. The precipitation anomalies are then used for subsequent statistical analysis to determine the two statistical metrices ($AR_1$ and s.d.). We clarified the meaning of quasi-stationary in the revision.

[Figure]

**Figure R1.** The statistics of precipitation in Miami (1948-2019): (a) the solid and dashed red line denotes the fitted lines before and after the critical transition year (solid black dot) (b) the two different solid red line are the mean values for each parts spilt by the critical transition year.

*7. Was the critical transition year 1994 identified a priori from the method described in point 6?*
Yes, the critical transition year(s) were all determined prior to statistical and network analysis, as illustrated in the response above.

*8. Line 326 – 332: how is the claim supported? Maybe an additional figure or if not important may choose not to mention. The reviewer is confused.*
This was only to show that our result is consistent to prior findings. We removed this part for better clarity.

*9. For city-scale analysis, is each transition year for each city identified using the same method as that for global scale?*

Yes, the same method (as illustrated in Fig. R1) was applied when identifying the transition year for each city.

*10. Why is CONUS PET not analyzed? Is it due to a lack of data?*
Right. The dataset we used for CONUS precipitation analysis does not contain PET data.

*11.Figure 4a, is it possible to indicate region number 1-9 corresponding to the adjacency matrix in a? This will facilitate the readers to connect the meaning of b to the spatial pattern of the network in a.*
Thanks for the advice. We replaced the original Fig. 4 with Fig. R2 shown below, where cities (nodes) are colored based on their corresponding climate region.

[Figure]

**Figure R2.** The precipitation network of CONUS cities: (a) the geographic map of connectivity and (b) the adjacency matrix, with $A_{ij} = 1$ in black (connected), $A_{ij} = 0$ in white, and red lines marking the division of nine geographic regions as shown in (a).

*12. The trend of AR1 within the moving window prior to transition year: in figures showing this metric, the non-monotonic trend can make it less useful as an indicator.*
In analyses of the behavior of real dynamic system, both AR1 and s.d. often exhibit non-monotonic trend, deviating from the theoretically increasing trends. This phenomenon has been consistently found in prior studies (e.g. Scheffer et al., 2009; C. Wang et al., 2020), and must involve the complex interactions of multiple determinants of the system (e.g. North Atlantic Oscillation, ENSO, and other low frequency variabilities for annal precipitation in CONUS). Because of this, caveats need to be taken using a singular indicator. It is also the very motivation behind this study that we look for more usable indicators (from statistical to network structure) so the critical transition can be more firmly determined by cross examine multiple indicators.

*13.Another question the reviewer is wondering about: I understand the paper's network analysis focuses on the network topology structure prior to transition, but will the network structure end up being different after the transition? i.e. will the enhanced network connectivity stay or*

*gradually 'relax' towards some 'climatological equilibrium state'?*
Thanks for the very insightful comment. We illustrate the trend of changes before and after the transition for CONUS precipitation, using s.d. and clustering coefficient (as they appear more reliable than other measures). The results are shown below in Fig. R3. Apparently, both trends relaxed after the transition. Yet, there are time lags (potential hysteresis) for different indicators (e.g. the clustering coefficient plateaued slightly after the transition year and gradually relaxed). This is somehow expected as the network parameters represented the "concatenated" system behavior, and should experience some lag in response and relaxation to the critical transition.

[Figure]

**Figure R3.** Two different metrices of CONUS precipitation: (a) the conventional s.d., and (b) the network clustering coefficient.

*14. The phrase potentially catastrophic transition may be less emphasized: the mean precipitation anomalies (for CONUS cities) are analyzed. If another variable like the maximum precipitation or number of days exceeding historical summer mean (just arbitrary examples of more catastrophic flavor), the reviewers will be more convinced.*

We agree that the phrase "catastrophic" is too strong for transitions in precipitation. We rephrased using "critical transition" throughout the manuscript.

In this study, we are focused on long-term (climatological) transition in precipitation (and PET), so annual means are good for this purpose. For maximum precipitation (or PET, or drought), it will be more natural to shift the focus to extreme events at meteorological scales. Theoretically, the concept of critical transition and the methodology developed in this study should be applicable. But practical difficulties will arise, such as the length of dataset (number of days for extreme precipitation) might be inadequate to discern the dynamic evolution of the network structure. Nevertheless, we have this in mind for our future research endeavor.

*15. The reviewer thinks that making more efforts to connect the global scale to the city scale will make the paper more coherent. For example, results in Fig. 3 are partially tied to global climates. Figure 5 and Figure 2 also seem to have some connections. In the introduction, the motivation for city-scale analysis may allude to some of these findings. E.g. city-scale responses are embedded in global hydrologic cycle changes but form systemic coherent structures/patterns – highly appealing to system-based network analysis.*

We thank the reviewer for this constructive comment. Yes, from the results of the study, we speculate that there is a positive correlation between the dynamics of precipitation in individual cities and regional/global trends, especially as we focus on the precipitation climatology. On the other hand, as we switch the spatial scale from local city to regional, the active determinants for precipitation are expected to change as well. For example, anthropogenic emissions of heat and aerosols are expected to have strong influence on local precipitation, whereas their impact on global scale might be diluted or replace by larger scale (and low-frequency) oscillations (e.g. ENSO). Future research along the line suggested by the reviewer will be promising, albeit we are refrained to make too strong assertion or speculation in the current study given the limited scope and results available at this stage.

---

## Author Comment (AC2)

**Response to the comments of Reviewer #2:**

*This paper focuses on critical transitions in precipitation and potential evapotranspiration (PET) both globally and in U.S. urban areas, based on monthly and annual datasets. The authors use various network and correlation measures to identify how system properties change leading up to critical transitions, which are defined as abrupt changes in behavior. They find that autocorrelation and standard deviation computed on moving time windows tend to increase before a defined critical transition point, indicating the potential use as early warning indicators. In an extension to a spatial network of precipitation in urban regions, the authors introduce network connectivity measures and similarly consider how these measures predict critical transitions in precipitation anomalies.*

*The paper was interesting and relevant to the journal and I think it will make a valuable contribution. The addition of a spatial network perspective on critical transitions in precipitation was particularly interesting to me. However, I have several major and minor comments on the structure of the paper and the methods as detailed below. Mainly, in the methods I would like to question the identification of a critical transition in general, and the possibility for trends in indicators without any critical transition occurring. In terms of the writing, there is a lot going on and several of the sections could be more clearly explained and tied together.*

*\*Note, after writing this initial review, I notice that some comments were addressed based on previous reviewer comments, but have not removed them, so they may now be redundant.*

We'd like to thank the reviewer for the constructive feedback and help in improving the quality of this manuscript. Below are detailed responses to the comments. All changes and clarifications were included in the revised manuscript and highlighted (yellow to Reviewer #1, and green Reviewer #2).

*Major comments:*

*Writing:*
*The introduction would benefit from some restructuring. For example, there are 3 separate places where different "research gaps" are established, and these could be better tied together Specifically: line 69 where "early warning signals remain obscure", line 83 where "hydrological processes remain un-explored", and line 87 "few studies have examined climate similarity". This makes it hard to follow what is actually being addressed. These could be combined into one more specific statement about what the literature has not fully addressed, that directly leads in to how you address it.*

Thanks for the comment. We have removed the line 69 and line 87 from the original manuscript. And we have rephrased line 83 to make the statement concise and easier to follow.

*From the methods, it is clear that many different datasets and metrics are used in this study, and some sort of illustrative figure or flow chart would be really useful here. For example, you use 3 different precipitation datasets at different scales, have a temporal analysis and a spatial analysis, yearly and monthly data, and several statistical measures. It would be good to have an overview of this at the beginning of the methods section (and/or a figure) to tie*

*these different parts of the study together.*

We did try to summarize the use of different datasets in a tabulated form in Section 2.1, and found it not much more informative than the current text summary with links to each dataset. We believe the tie of subsequent section of results to the corresponding dataset is made self-clear when we refer to results of "global", "regional", or "city" scale, respectively. We are open to have an additional table for dataset summary if the reviewer finds it more convenient.

*Figure 1: I like that you have included an illustrative example, but it comes very suddenly (I was surprised by "harvest" and thought it was somehow linked to precipitation) and is not fully explained. This example could use its own subsection and then some linkages to exactly what we are looking for in the following precipitation-based results.*

Thanks for the comments. As same with the feedback provided by the reviewer #1, the harvest model is a *benchmark* problem used to illustrate the concept of *early warning* signals of critical transition, particularly the increasing trends of s.d. and $AR_1$. We added some transitional phrase in the context in the hope that it will not come up as a "surprise".

*Related to the above, the introduction and methods section seem a lot longer than the actual results and discussion of the study. The results section would benefit from more discussion, and ties between sections. For example, many studies are brought up in the introduction, and some could be moved here to compare with your specific results. Also since the methods are heavy on different metrics, the reader could use reminders of what some of these metrics mean from a physical standpoint within the results.*

The methods section is heavy because the major novelty of the current study is the introduction of new network-based metrics, while the results turned out to be a natural "proof-of-concept" and did not involve extensive discussion.
Per your suggestion, we removed some studies in the introduction into the results section to make the whole structure more balanced. For the network-based metrics, we added concise reminders of the physical meanings of them in the context to make the reading smoother.

*Methods and interpretation:*
*I have a question on the selection of a critical transition for a given time-series: Can there be a time-series with no critical transition? Currently I get the idea that this time point is selected in every dataset as the maximum rate of change, or "abrupt change of slopes"...which does not necessarily indicate a critical transition, but adjacent years with high variability.  I thought it would make more sense to define a critical transition as a step change, where magnitudes or statistical properties "before" and "after" are maximally different. In general, the definition and reasoning to identify a critical transition should be more clear. As it is, referring to changes in precip between two years as a "catastrophic transition" seems tenuous.*

Firstly, yes, there are time-series with no critical transition. But then the time series will be of no use to our purpose of illustrating early-warning signals of critical transitions; thus they are naturally excluded.

The process of detecting "critical transitions" in time series of observational datasets is illustrated below in Fig. R1, using the precipitation climatology of the city of Miami as an

example, where the year of transition is found to be in 1998. Other critical transition detection follows exactly the same procedure. As will be made clear, the "catastrophic transitions", though it is determined in a specific year, does not refer to changes in precipitation between two particular **years** before and after this point (i.e. Year 1998 in Miami), but really to the transition of the *longterm historical evolution* of precipitation climatology prior to that particular year into a new trend of precipitation pattern after it.

To begin with, we bisected the cumulative precipitation data (scatters) and fitted each segment using linear regression (thus each being *quasi-stationary* with a constant slope). The year of transition is determined as the intersection of two trend lines (solid red: prior to transition and dashed red: after transition) with different slopes. In addition, Fig.R1b shows the annual (not cumulative) precipitation, where the two different means (prior to and after the transition year) are subtracted. The precipitation anomalies are then used for subsequent statistical analysis to determine the two statistical metrics ($AR_1$ and s.d.). We clarified the meaning of quasi-stationary in the revision.

[Figure]

**Figure R1.** The statistics of precipitation in Miami (1948-2019): (a) the solid and dashed red line denote the fitted lines before and after the critical transition year (solid black dot) (b) the two different solid red lines are the mean values for each part spilt by the critical transition year.

*The time windows (or data lengths) over which statistical measures are calculated is very small, e.g. 13 or 7 annual data points in all these cases. This leads to some question of statistical significance of trends in the early warning indicators. For example, you highlight several cities out of a much larger pool of data that show these increases in AR and stdev before a critical transition, but is that actually typical? Or are there many cases where these indicators are increasing where there is no critical transition (false positives), or cases where they do not well predict a transition (false negatives)?*

We tested the size of moving windows continuously between 5 to 25 years as suggested by a prior climatology studies from Tsonis et al. (2007). And the window sizes of 13 years (for the global dataset) and 7 years (for the CONUS dataset) were determined because these window sizes yield the most statistically manifest results, while other window sizes give similar trends of evolution of early-warning signals but not as manifest. The same procedure was performed in an earlier work (Wang et al., 2020) and was proven using more rigorous

statistical test, such as sensitivity of Kendall's $\tau$ to window sizes, which we did carried out in this study but did not report. See Fig. 2 in
Wang, C., Wang, Z.H., & Sun, L. (2020). Early warning signals for critical temperature transition. Geophysical Research Letters, 47, e2020GL088503.

It is a very insightful question if "are there many cases where these indicators are increasing where there is no critical transition (false positives), or cases where they do not well predict a transition (false negatives)". To illustrate, we plotted the trend of changes before and after the transition for CONUS precipitation, using s.d. and clustering coefficient (as they appear more reliable than other measures). The results are shown below in Fig. R2. There does not seem to be any *false positives*, but the identified critical transitions are the only manifest positives over this sufficiently long period of time, if we consider the global trends (not the local crests or troughs). In addition, both trends relaxed after the transitions, and there does not seem to be *positive negatives* neither. Yet, there are time lags (potential hysteresis) for different indicators (e.g. the clustering coefficient plateaued slightly after the transition year and gradually relaxed). This is somehow expected as the network parameters represented the "concatenated" system behavior, and should experience some lag in response and relaxation to the critical transition. We really appreciate your insightful comments and hope this clarifies.

[Figure]

**Figure R2.** Two different metrices of CONUS precipitation: (a) the conventional s.d., and (b) the network clustering coefficient.

*I liked the network analysis, but it was hard to go between the table of the regions and Figure 4 – could the table with the regions be made into a colored map that goes into Figure 4? This would tie these regions into the results in more directly and make them easier to discuss.*

Thanks for the advice. As it was also suggested by the other reviewer, we have modified the network map and incorporated different regions corresponding to the table with different colors. The revised Fig. 4 is shown below in Fig. R3. We have revised it in the revision.

[Figure]

**Figure R3.** The precipitation network of CONUS cities: (a) The geographic map of connectivity and (b) the adjacency matrix, with $A_{ij}=1$ in black (connected), $A_{ij}=0$ in white, and red lines marking the division of nine geographic regions as shown in (a)

*Minor comments:*

*Line 7: The first sentence of the abstract could be restructured to not start with "In this study…" as that is apparent.*
We rephrase the first sentence as suggested.

*Line 17: shed new light*
Typo corrected.

*Line 55: get rid of "aka" and explain fully. Similarly, in various places, recommend getting rid of term "viz" and explaining fully.*
We replaced the "aka" and all other "viz" with the full expressions.

*Line 84: cites = cities*
Typo corrected.

*Line 100: I don't think PET has been defined, or could use re-defining here*
We defined potential evapotranspiration (PET) as it first appears in line 97.

*For the AR1 as a measure (e.g. on the y-axis of several figures) – is the measure itself actually the alpha term in Equation 1? Or This was not completely clear to me at first, since the label is just "AR1". Actually, it seems like Equation 1 lines up with Equation 4, and 3 goes with 5, so perhaps this subsection could be better re-organized and less repetitive.*
The alpha term in Equation 1 is the autocorrelation coefficient in the simple autoregression model, whereas the lag-1 autocorrelation AR1 is defined in Equation 4. We now deleted the name of the alpha term in Eq. 1 as the autocorrelation as it causes confusion with AR1, and the alpha term is an intermediate variable which was not used again in subsequent sections.

Hope this helps to make better clarity.

*Line 172: governing dynamics are*
Typo corrected.

*Line 188: another statement of a research gap, not needed here really*
We removed this sentence to avoid the repetition.

*Line 189: emerge*
The sentence was removed.

*Line 205: measure*
Corrected.

*Line 265: You had over 100 cities in this analysis according to the methods but only introduce and discuss these 4, would be good to rationalize that small election (as they are exemplary, show the largest trends in early warning indicators, etc).*
We pre-selected these 4 cities out of all 481 CONUS cities/towns as representative to their distinct geographic and climatic conditions, *before* we carried out the actual early-warning signal analysis. To verify, we conducted the same analysis to other cities: either they exhibit similar indicators when transitions exist, or no manifest signals detected. The first case was dismissed as repetitive, and the second due to insignificance. We added rationale to explain the choice in the revision.

*Line 288: "highly assortative with large modularity" needs more expansion*
We added physical explanation to this phrase in the context.

*Line 297: responds, presages*
Typos corrected.

---

## Author Response (AR2)

**Response to the comments of Reviewer #2:**

We'd like to thank the reviewer for the constructive feedback and help in improving the quality of this manuscript. Below are detailed responses to the comments. All changes and clarifications are included in the revised manuscript and highlighted

*Thanks to the authors for their clear responses to my previous comments and suggestions. I think the paper is significantly improved, and now have a short set of comments below. Otherwise, I think this is an interesting paper and worthy of publication with minor revisions.*

*Major comment:*
*My initial comment regarding the detailed references to studies in the introduction seeming separate from the results and discussion of the study remains. In general, I think it would be valuable to discuss more findings related to precipitation shifts in the context of your actual results and make connections between the scales you analyze. This also goes along with a previous comment from Reviewer 1, who notes several connections that could be made between the cities versus regional trends, and notes that some of the content from the introduction could be useful. If the authors do not want to change this aspect, I would ask for a stronger rationale for such detailed introduction and methods sections and a relatively very brief exposition of the study findings. As it is, I felt like the paper was nearly over after the methods were described.*

We thank the reviewer's insight in improving the overall structure of the paper, especially the seeming gap between the introduction (Section 1) and results and discussion (Section 3). In this revision, we strengthened the link by explicitly referencing Section 3 in the end of the introduction and added the justification of using different spatial (city, regional, and global) scales in Sections 3.1 and 3.2. See also our response below for detailed revision.

*Minor comments:*

*Line 95: I think this section, or somewhere in the introduction, could use a little more detail on the actual study – for example mention the different scales analyzed and the reasoning for these scales (which can tie in to the background you have already provided).*

This part is substantively revised as follows:

"In this paper, we aim to investigate critical transitions in hydrological processes, primarily precipitation, at various spatial, ranging from city to global, scales, using both conventional statistical and novel network measures. Detailed analyses at different scales demonstrate the versatility of the proposed method and will be of interest to locality-concerned researchers and policy makers. In particular, the analysis in individual U.S. cities (see Section 3.1 and Fig. 3 below) will enhance our understanding of the physics of urban hydroclimate via local land-atmosphere interactions (Song and Wang, 2015, 2016).

The remainder the paper is organized as follows: we present data sources of precipitation and potential evapotranspiration (PET) in Section 2, together with definition of early-warning signals and basic network analysis techniques. These methods are then applied to urban areas in CONUS with results presented in Section 3: statistical variance and autocorrelation in Section 3.1, and changes in network structure in Section 3.2. Specifically, the results of

Section 3.1 are on PET analysis at the global scale and the precipitation climatology at city and global scales. The network analysis in Section 3.2 is applied to the regional precipitation in CONUS. The choice of different scales in Section 3 is partly due to data availability (such as the inadequacy of precipitation data to construct precipitation network at individual city or global scales), and partly to avoid repetition of similar findings (such as the trend of statistical variance and autocorrelation of the CONUS precipitation closely resembles its urban-scale counterparts). We then conclude this study with main findings and future perspectives in Section 4."

*Line 174: The authors noted that they added the phrase "benchmark example" to make the harvesting example more clear here. However, I think this really deserves its own subsection (e.g. "Illustrative Example" to make it a separate part of the methods section that prepares the reader to understand the actual results.*

We made this part a separate subsection 2.2.3 titled "Illustrative example of the autocorrelation process".

*Section 2.2.2: "Statistical" instead of statistic*

Typo corrected.

*Code/Data availability: I would recommend a stronger statement on data availability, as in general codes should be made available in a repository. However, this is a journal requirement so maybe this is fine for HESS.*

Here we followed the standard language of the journal for code/data availability. We will make the code/metadata available through online data repository as mandated by our funding agencies.